



# Technical Note: A software framework for calculating compositionally dependent *in situ* [14]C production rates

Alexandria J. Koester[1], Nathaniel A. Lifton[1,2]

[1]Department of Earth, Atmospheric, and Planetary Sciences, Purdue University, West Lafayette, IN 47907, USA
[2]Department of Physics and Astronomy, Purdue University, West Lafayette, IN 47907, USA

*Correspondence to*: Alexandria J. Koester (koestea@purdue.edu)

**Abstract**

Over the last 30 years, *in situ* cosmogenic nuclides (CNs) have revolutionized surficial process and Quaternary geologic
studies. Commonly measured CNs extracted from the common mineral quartz have long half-lives (e.g., [10]Be, [26]Al), and have
been applied over timescales from a few hundred years to millions of years. However, their long half-lives also render them
largely insensitive to complex histories of burial and exposure less than ca. 100 ky. On the other hand, *in situ* cosmogenic [14]C (*in
situ* [14]C) is also produced in quartz, yet its 5.7 ky half-life renders it very sensitive to complex exposure histories during the last
~25 ka – a particularly unique and powerful tool when analyzed in concert with long-lived nuclides. *In situ* [14]C measurements
are currently limited to relatively coarse-grained (typically sand-sized or larger, crushed/sieved to sand) quartz-bearing rock
types, but while such rocks are common, they are not ubiquitous. The ability to extract and interpret *in situ* [14]C from quartz-poor
and fine-grained rocks would thus open its unique applications to a broader array of landscape elements and environments.

As a first step toward this goal, a robust means of interpreting *in situ* [14]C concentrations derived from rocks and minerals
spanning wider compositional and textural ranges will be crucial. We have thus developed a MATLAB®-based software
framework to quantify spallogenic production of *in situ* [14]C from a broad range of silicate rock and mineral compositions,
including rocks too fine-grained to achieve pure quartz separates. As expected from prior work, production from oxygen
dominates the overall *in situ* [14]C signal, accounting for >90% of production for common silicate minerals and six different rock
types at sea-level and high latitudes (SLHL). This work confirms that Si, Al, and Mg are important targets, but also predicts
greater production from Na than from those targets. The compositionally dependent production rates for rock and mineral
compositions investigated here are typically lower than that of quartz, although that predicted for albite is comparable to quartz,
reflecting the significance of production from Na. Predicted production rates drop as compositions become more mafic
(particularly Fe-rich). This framework should thus be a useful tool in efforts to broaden the utility of *in situ* [14]C to quartz-poor
and fine-grained rock types, but future improvements in measured and modelled excitation functions would be beneficial.

## 1 Introduction

Rare nuclides produced *in situ* in minerals near the Earth's surface by cosmic-ray bombardment (*in situ* cosmogenic nuclides
or CNs) have revolutionized studies of geomorphology and Quaternary geology. CNs build predictably over time in an exposed
surface through nucleon spallation and muon reactions (e.g., Gosse and Phillips, 2001). As such, the time at which geomorphic
surfaces formed by glacial, fluvial, or marine activity often can be constrained with CNs, an application known as surface
exposure dating. In addition, CNs can be used to constrain rates of surficial processes with appropriate interpretive models.
These applications rely on measuring the concentrations (atoms g[-1]) of CNs in a sample and calculating an exposure age or





erosion rate based on the production rate (atoms g$^{-1}$ yr$^{-1}$). The most-commonly measured CNs, $^{10}$Be and $^{26}$Al (t$_{1/2}$ 1.39 My - Korschinek et al. (2010); Chmeleff et al. (2010); and t$_{1/2}$ 0.705 My - Nishiizumi et al. (2004), respectively), are typically extracted from quartz, due to its simple composition and corresponding resistance to weathering under a wide range of environmental conditions. Their long half-lives make these nuclides useful in dating surfaces that have been exposed up to

millions of years. However, their half-lives also render their concentrations insensitive to periods of burial and re-exposure of less than ca. 100 ky – this can lead to problems with exposure dating due to nuclide inventories remaining from prior periods of exposure.

*In situ* cosmogenic $^{14}$C (*in situ* $^{14}$C) is also produced in quartz, but its 5.7 ky half-life limits its utility for simple exposure dating because its concentration reaches secular equilibrium between production and decay after 25-30 ky of continuous

exposure. However, its rapid decay also makes it sensitive to complex periods of burial and exposure since ca. 25-30 ka (e.g., Briner et al., 2014). In addition, its short half-life means measured concentrations are sensitive only to very rapid erosion rates (e.g., Gosse and Phillips, 2001; von Blanckenburg et al., 2005; Hippe et al., 2017; Hippe et al., 2021), making many eroding landscape elements good targets for *in situ* $^{14}$C studies. *In situ* $^{14}$C is thus emerging as a powerful addition to the CN toolkit.

Several techniques for extracting *in situ* $^{14}$C from sand-sized quartz grains have been established (Lifton et al., 2001; Lifton

et al., 2015; Goehring et al., 2019; Hippe et al., 2013; Lupker et al., 2019; Fülöp et al., 2019), but while coarse-grained quartz is common, it is not ubiquitous. Landscapes dominated by mafic or intermediate lithologies generally lack quartz, and fine-grained lithologies can limit the efficacy of quartz purification techniques, thus applying *in situ* $^{14}$C to such rock types is currently problematic. However, the ability to extract and interpret *in situ* $^{14}$C concentrations reliably from quartz-poor and fine-grained lithologies would significantly broaden its applications to additional landscapes and enable pairing with additional nuclides such

as $^{36}$Cl. Indeed, early studies of *in situ* $^{14}$C in terrestrial rocks utilized whole-rock samples (e.g., Jull et al., 1992; 1994), until procedural difficulties shifted the focus to the simpler quartz production and extraction systematics (Lifton, 1997; Lifton et al., 2001).

As a first step in expanding the range of available sample targets, we have developed a software framework that estimates the production of *in situ* $^{14}$C from major elements found in typical rocks and potential mineral separates. We modified the

MATLAB® code from Lifton et al. (2014) to calculate compositionally dependent, site-specific production rates using nuclide-specific scaling, major-element oxide compositions, and measured and modelled nucleon excitation functions, referenced to geologically calibrated *in situ* $^{14}$C spallogenic production rates in quartz. This new framework thus provides a critical first step for potential future applications incorporating quartz-poor or fine-grained samples.

## 2 Constraining compositionally dependent *in situ* $^{14}$C production rates

### 2.1 Geologic and experimental production rate calibrations

*In situ* CN applications require accurate estimates of the rate at which a given nuclide of interest is produced in the target mineral or rock. This is typically achieved by calibrating the production rate with CN measurements in samples from one or more sites with 1) an independently well-constrained exposure history (e.g., Borchers et al., 2016; Phillips et al., 2016; Lifton et al., 2015a), or for radionuclides only, with 2) demonstrable surface stability such that measured CN concentrations can be

inferred to have reached a secular equilibrium between production and decay, at which point the concentration is only a function of time-integrated production rate and the decay constant (e.g., Jull et al., 1992; Borchers et al., 2016). Production rates can also





be calibrated experimentally by exposing high-purity, low background targets to the secondary cosmic-ray flux at given sites for a known duration under well-constrained conditions (e.g., Nishiizumi et al., 1996; Brown et al., 2000; Vermeesch et al, 2009).

Since production rates cannot be calibrated at every place on Earth, these site-specific estimates are typically scaled to other sites of interest using an appropriate scaling framework that accounts for spatial and temporal variations in the secondary cosmic-ray flux, arising from fluctuations in the geomagnetic field (parameterized by effective vertical cutoff rigidity, $R_C$, in GV), atmospheric depth ($X$, in g cm$^{-2}$), and solar modulation (described by the parameter $\Phi$, in MeV) (e.g., Lifton et al., 2014). Such scaling frameworks are typically referenced to conditions corresponding to sea-level and high geomagnetic latitude (SLHL).

Geologic calibrations are generally preferable for minerals with specific compositions since samples from well-constrained sites should incorporate natural variability relevant over geologic time spans. Such calibrations for *in situ* $^{14}$C have focused on quartz to date, given its simple chemistry and weathering resistance (e.g., Borchers et al., 2016; Phillips et al., 2016; Lifton et al., 2015a; Schimmelpfennig et al., 2012; Young et al., 2014), yet variable compositions require more complicated consideration of the compositional dependence of CN production (e.g., $^{36}$Cl; Marrero et al., 2016a). It is often useful in such cases to utilize theoretical production rate estimates based on integrals of the differential cosmic-ray flux and the relationship between reaction probability and incident particle energy.

### 2.2 Theoretical production rate estimates

The probability that a given nuclear reaction will occur at a given kinetic energy $E$ of an incident particle is described by the reaction cross-section ($\sigma$), in units of barns (1 barn = $10^{-24}$ cm$^2$). With the advent of accelerator mass spectrometry (AMS), cross-

section measurements for reactions producing CNs have become relatively common, and knowledge of the variation of $\sigma$ as a function of $E$ for those reactions (known as an excitation function) are continuing to improve (e.g., Reedy, 2013). Proton-induced reactions are simpler to measure than those induced by neutrons because it is easier to accelerate protons into a mono-energetic beam. Mono-energetic (or quasi-mono-energetic) neutron reaction cross-sections are more difficult to obtain, however, and thus are often estimated from analogous proton cross-sections (Reedy, 2013).

Measured or modelled excitation functions can then be used to estimate theoretical production rates for a CN of interest using Eq. (1) below (e.g., Masarik and Beer, 2009),

$$P_j(X, R_C, \Phi) = \sum_i ND_i \sum_k \int_0^\infty \sigma_{ijk}(E_k) J_k(E_k, X, R_C, \Phi) \, dE_k \qquad (1)$$

where $ND_i$ is the target number density, or number of atoms of the target element $i$ per gram of sample material (atoms g$^{-1}$), $\sigma_{ijk}(E_k)$ is the cross-section for the production of nuclide $j$ (cm$^2$) by particles of type $k$ with energy $E_k$ (MeV), and $J_k$ ($E_k$, $X$, $R_C$,

$\Phi$) is the differential flux of atmospheric cosmic-ray particles (cm$^{-2}$ yr$^{-1}$ MeV$^{-1}$) of type $k$ with energy $E_k$ at a location and time specified by $X$, $R_C$, and $\Phi$.

The production of *in situ* $^{14}$C in silicates is dominantly from spallation of O, and theoretical simulations suggest minor spallogenic production from Mg, Al, and Si (Masarik and Reedy, 1995; Masarik, 2002). Production of *in situ* $^{14}$C from muons also occurs, either via slow negative muon capture or by fast muon interactions (Heisinger et al., 2002a,b). The muogenic

component of *in situ* $^{14}$C production in surficial quartz at SLHL is significant – on the order of 20% of total production (e.g., Lupker et al., 2015; Balco, 2017). However, muogenic production of *in situ* $^{14}$C has only been estimated experimentally from





$^{16}$O (Heisinger et al., 2002a; 2002b). Further work is needed in this area to better understand production from other muogenic reactions. We therefore focus on the dominant spallogenic pathways for the purposes of this initial study.

## 3 Methods

### 3.1 Software framework

Our MATLAB®-based compositionally dependent *in situ* $^{14}$C production rate software framework builds on the LSDn nuclide-dependent scaling formulation of Lifton et al. (2014), which uses the PARMA analytical approximations to Monte Carlo calculations of atmospheric differential flux spectra of neutrons, protons, and muons as a function of $X$, $R_C$, and $\Phi$ (Sato et al., 2006; 2008). We also incorporate the gridded $R_C$ and dipolar $R_{CD}$ models of Lifton et al. (2016), based on the SHA.DIF.14k

paleomagnetic model (Pavón-Carrasco et al., 2014). This work accounts for effects of variable sample compositions on *in situ* $^{14}$C production by incorporating relevant reaction excitation functions and number densities for elements in the standard suite of major-element oxide compositions. Output from this new framework should complement current web-based cosmogenic nuclide calculators incorporating the LSDn scaling framework and *in situ* $^{14}$C, including version 3 of the University of Washington cosmogenic-nuclide calculators (herein UWv3: hess.ess.washington.edu) (Balco et al., 2008) and the Cosmic-Ray-prOduced

NUclide Systematics on Earth project (CRONUS-Earth) calculator (CRONUSCalc; http://cronus.cosmogenicnuclides.rocks/; Marrero et al., 2016b).

Reaction excitation functions for neutrons and protons were compiled from Reedy (2007; 2013), and the JENDL/HE-2007 database (Fukahori et al., 2002; Watanabe et al., 2011) found in the online Evaluated Nuclear Data File (ENDF, https://www-nds.iaea.org/exfor/endf.htm, accessed April 2020; Brown et al., 2018) for each of the major elements included in typical

elemental oxide analyses. We consider empirical excitation functions to be generally more reliable than those derived from nuclear reaction models, and thus use measured functions if available. Five neutron and proton excitation functions are based on measurements from Reedy (2007, 2013) ($^{16}$O, $^{24}$Mg, $^{27}$Al, $^{28}$Si, $^{56}$Fe) while we used modelled neutron and proton reaction excitation functions from JENDL/HE-2007 for the remaining elements ($^{23}$Na, $^{31}$P, $^{39}$K, $^{40}$Ca, $^{48}$Ti, $^{55}$Mn). We utilized the JENDL/HE-2007 database because the relevant excitation functions extended to a maximum energy of 3 GeV. The exceptions

were the excitation functions for $^{31}$P, extending only to 0.2 GeV. Each excitation function was interpolated into logarithmic energy bins from 1 MeV to 200 GeV for both neutron (XX(n,x)$^{14}$C) and proton (XX(p,x)$^{14}$C) reactions, where XX is the target nuclide (Fig. 1). The cross-section at the highest measured or modelled energy reported for each excitation function is assumed to be constant beyond that energy up to 200 GeV, the maximum energy we consider.

We incorporate sample compositions using common major elemental oxide analyses (e.g., from X-Ray Fluorescence (XRF)

measurements) to calculate ND for each element considered in Eq. 1. The ND value for each target element in a sample is then calculated per Eq. (2), for input to Eq. 1:

$$ND = \frac{E_{Fr}*E_{Ox}*N_A}{100*A_m},\qquad(2)$$

where $E_{Fr}$ is the elemental fraction in each oxide (formula mass of each element in its oxide divided by the total formula mass of the oxide (e.g., Mg/MgO or 2Al/Al$_2$O$_3$)), $E_{Ox}$ is the measured major elemental oxide weight percent input by the user, $N_A$ is

Avogadro's number (6.02214076 x $10^{23}$ atoms mol$^{-1}$) and $A_m$ is the molar mass of the element in g.



### 3.2 Predicted compositionally dependent production rates

Theoretical compositionally dependent site-specific *in situ* [14]C production rates are reported relative to the SLHL *in situ* [14]C production rate in quartz, geologically calibrated as part of the CRONUS-Earth project (e.g., Phillips et al., 2016; Borchers et al, 2016) and supplemented with two subsequent production rate calibration datasets (Schimmelpfennig et al., 2012; Young et al., 2014), using the LSDn scaling framework (Lifton et al., 2014, Lifton 2016). SLHL estimates are referenced to the year 2010 (Lifton et al., 2014; Lifton, 2016) assuming an atmospheric pressure of 1013.25 hPa (converted to atmospheric depth, g cm$^{-2}$), an $R_c$ value of 0 GV, a Φ value of 624.5718 MV, and a fractional water content value, '$w$', of 0.066 (Sato et al., 2006; Phillips et al. 2016). We recalibrated the *in situ* [14]C spallogenic production rate at SLHL in quartz from the studies above by first calculating the unweighted mean and standard deviation of replicate analyses of samples at each site (to avoid biasing the results toward sites with more analyses). Best-fitting SLHL production rate estimates for each site were determined using a χ$^2$ minimization procedure. The unweighted mean and standard deviation of all sites were then calculated from the site-specific SLHL production rate estimates, yielding global SLHL values for quartz of 13.5 ± 0.9 atoms g$^{-1}$ yr$^{-1}$ and 13.7 ± 1.2 atoms g$^{-1}$ yr$^{-1}$ for the gridded $R_C$ and geocentric dipolar $R_{CD}$ records of Lifton (2016), respectively, as noted above. The latter is comparable to the calibrated value generated by the UWv3 calculator from the same dataset. In the following discussion we focus on the gridded $R_C$ value (referenced below as $P_{Qcal}$), as it provides a somewhat better fit to the global calibration dataset. Corresponding geocentric dipolar values are included in the Supplement.

For comparison, the purely theoretical *in situ* [14]C production rate by nucleon spallation predicted at SLHL in quartz using Eq. 1 is 15.8 atoms g$^{-1}$ yr$^{-1}$ ($P_{Qref}$). This discrepancy with the calibrated value likely reflects uncertainties in both the excitation functions and the nucleon fluxes considered (Reedy, 2013; Sato et al., 2006; Sato et al., 2008). Giving more credence to the geologically calibrated quartz values, we account for this discrepancy similarly to Lifton et al. (2014), deriving a compositionally dependent site-specific production rate ($P_{CD}$) by normalizing the predicted compositionally dependent production rate at the site of interest ($P_{CDpred}$) by the ratio of $P_{Qcal}$ to $P_{Qref}$, per Eq. 3. Another way to think of this is that the ratio of $P_{CDpred}$ to $P_{Qref}$ is the compositionally dependent scaling factor, multiplied by the geologically calibrated production rate in quartz, $P_{Qcal}$.

$$P_{CD} = P_{Qcal} \frac{P_{CDpred}}{P_{Qref}} \quad \text{atoms g}^{-1} \text{ yr}^{-1} \tag{3}$$

We compare $P_{CD}$ values at SLHL to $P_{Qcal}$ for compositions reflecting both individual minerals (Barthelmy, 2014) (i.e., mineral separates) and a broad range of silicate rock types (Parker, 1967; Fabryka-Martin, 1988) (i.e., whole-rock analyses) (Table 1). A pure calcite composition ($CaCO_3$) is assumed for limestone and $MgCa(CO_3)_2$ is assumed for dolomite. Spallation production is only possible from Ca and O, although we included the O number density contribution from $CO_2$ in the software framework. Thermal neutron production of *in situ* [14]C from [12]C or [13]C is expected to be negligible and is not considered here (e.g., Wright e al., 2019).

### 4 Results and Discussion

#### 4.1 Predicted modern production rates for silicate minerals and rock types

Predicted SLHL modern (i.e., 2010) spallogenic production rates for *in situ* [14]C in the silicates considered here are generally lower than that from pure quartz (Table 2), but spallation production from [16]O dominates throughout the compositional range we





explored (Table 3). As expected from reaction systematics, $^{14}$C production rates tend to decline rapidly with progressively increasing atomic mass of the target nuclide. Interestingly, the production rate predicted for albite using the excitation functions from JENDL/HE-2007 for spallation reactions on $^{23}$Na is comparable to that of quartz. We note that the JENDL/HE-2007 model $^{23}$Na(n,x)$^{14}$C excitation function exhibits a broad peak between ca. 30-350 MeV with cross-sections comparable to that of the

empirical $^{16}$O(n,x)$^{14}$C excitation function of Reedy (2013) (Fig. 1), suggesting similar production magnitudes for the two reactions. To our knowledge, no comparable empirical excitation functions for the $^{23}$Na(n,x)$^{14}$C or $^{23}$Na(p,x)$^{14}$C reactions have been published to date, making the model reactions difficult to validate. Predicted production rates for Mg-rich silicates such as forsterite and enstatite are ca. 7-10% lower than in quartz, while Al-rich minerals such as Ca- and K-feldspars yield production rates 12-13% below quartz. Ca-rich wollastonite exhibits less than 1% of its total $^{14}$C production from Ca, yielding a production

rate more than 20% below that of quartz, while Fe-rich minerals such as ferrosilite and fayalite suggest SLHL production rates ca. 32% and 41% less than quartz, respectively. Predicted production rates for two carbonate minerals considered, calcite and dolomite, are 12% and 3% less than quartz, respectively.

The $P_{CD}$ values for selected rock types (ultramafic, basalt, high-Ca granite, low-Ca granite, and granodiorite; Fabryka-Martin, 1988) follow a similar pattern to the individual minerals, with total production rates less than that of quartz but with less

overall variation (Table 2). Predicted whole-rock production rates tend to increase with decreasing Fe and Mg content, with $P_{CD}$ values ranging from nearly 15% less than quartz for ultramafic compositions to ca. 5-7% below that of quartz for more felsic compositions. As with the idealized mineral compositions, spallation from $^{16}$O dominates *in situ* $^{14}$C production (>90% for all compositions considered), with lesser production from Si, Al, Na, and Mg. Only minor production contributions from Ca and Fe are predicted (typically <1%).

**4.2 Assessing uncertainty in predicted compositionally dependent production rates**

There are three main sources of uncertainty in our predicted production rates, associated with the particle spectra, the geologic production rate calibration for *in situ* $^{14}$C in quartz, and the excitation functions. We note that these are not entirely independent, as the LSDn-based production rate calibration utilizes both the particle spectra of Sato et al. (2008) and excitation functions of Reedy (2013). Sato et al. (2008) quote statistical uncertainties in their modelled particle fluxes on the order of 5-

20% between ca. 10 km altitudes and sea level, respectively, although Lifton et al. (2014) note that predictions within this altitude range show good agreement with measured differential fluxes and no evidence of systematic errors. The conservative uncertainty in the recalibrated *in situ* $^{14}$C global production rate in quartz is on the order of 6-7% using the gridded $R_C$ geomagnetic framework and LSDn scaling. Reedy (2013) suggests uncertainties on the order of 10% for the empirical excitation functions presented. However, assessing the uncertainty in the modelled functions of JENDL/HE-2007 is more difficult.

We attempted to assess this latter uncertainty by comparing results using JENDL/HE-2007 to predictions incorporating the more recent TENDL-2019 database (Koning et al., 2019). We focused on the proton and neutron excitation functions for $^{14}$C production from $^{23}$Na, since our predictions using the JENDL/HE-2007 $^{23}$Na excitation functions suggest comparable production to that from $^{16}$O (Fig. 1; Table 2). However, TENDL-2019 excitation functions only extend to an energy of 200 MeV, although at higher resolution than JENDL/HE-2007. We thus compared albite production rates predicted using the JENDL/HE-2007

excitation function alone (Na$_J$) with those incorporating spliced neutron and proton excitation functions using TENDL-2019 for E ≤ 200 MeV and JENDL/HE-2007 for E > 200 MeV (Na$_{TJ}$) (Fig. 2).





Neutron and proton excitation functions for $^{23}$Na have similar thresholds of ca. 30-35 MeV in both JENDL/HE-2007 and TENDL-2019 (Fig. 2). Of note, the low-energy peaks in the TENDL-2019 excitation functions are narrower, ca. 30% lower, and occur at a slightly higher energy than those of JENDL/HE-2007 (ca. 150 MeV vs. ca. 90 MeV, respectively). However, the
predicted production rate for albite using the spliced $Na_{TJ}$ excitation functions is only ca. 3% less than that using the $Na_J$ excitation functions alone (Table 2); also reflected in the lower production proportion from Na of ca. 8% in the spliced version, vs. ca. 13% in $Na_J$ version (Table 3). Based on these results, we suggest assuming a 10% uncertainty as well for the JENDL/HE-2007 excitation functions overall, pending empirical validation. Thus, considering the three sources of uncertainty, we suggest a reasonable estimate of uncertainty on our theoretical production rates might be on the order of 10-15%.

**4.3 Comparisons with previous studies**

We compare output of our software framework to two earlier studies that also calculated theoretical *in situ* $^{14}$C production rates from targets of varying composition (Fabryka-Martin, 1988; Masarik, 2002), without adjusting our predictions to the geologically calibrated production rate in quartz. First, Fabryka-Martin (1988) estimated SLHL secular equilibrium *in situ* $^{14}$C concentrations at depths of ~20 cm for ultramafic rock, basalt, high-Ca granite, low-Ca granite, and limestone compositions,
following Parker (1967) (Table 4). The equilibrium concentrations were calculated assuming neutron spallation production only from oxygen and a SLHL production rate of 26 atoms g$^{-1}$ yr$^{-1}$ from oxygen (Yokoyama et al., 1977) based on excitation functions from Reedy and Arnold (1972). We derived secular equilibrium SLHL production rates from Fabryka-Martin (1988) by multiplying the concentrations by the $^{14}$C decay constant of 1.216 x 10$^{-4}$ y$^{-1}$ (Table 4 – $P_{16O\text{-}FM}$). Considering only theoretical production from $^{16}$O in our results (Total $P_{CDpred}$ in Table 2 multiplied by the corresponding O production proportion in Table 3),
our $P_{16O}$ values in Table 4 are ca. 40-45% below those derived from Fabryka-Martin (1988). However, it should be pointed out that Yokoyama et al. (1977) suggest ±35% uncertainty (1σ) on their *in situ* $^{14}$C production rate estimate used by Fabryka-Martin (1988), so our theoretical $P_{16O}$ values using more accurate particle fluxes and excitation functions lie well within that range.

The second study we considered (Masarik, 2002) is a conference abstract that presents formulas for estimating compositional dependence of *in situ* cosmogenic nuclide SLHL production rates by neutron spallation, including $^{14}$C, derived
from numerical simulations. For *in situ* $^{14}$C production, Masarik (2002) considers the target elements O, Mg, Al, Si, and Fe, parameterized in terms of weight fractions of each (Table 5). Total production rates from Masarik (2002) ($P_{M02}$) in Table 5 are typically ca. 10-20% higher than neutron-only theoretical production rates for rock and mineral compositions considered in this study (Neutron $P_{CDpred}$, Table 2). Being an abstract, details underlying the simulations and calculations in Masarik (2002) are sparse, but we suggest a combination of differences in the differential neutron flux spectra (Masarik and Beer, 1999, vs. Sato et
al., 2008) and excitation functions (e.g., Reedy and Masarik, 1995, vs. Reedy, 2013) used in the two studies may be the sources of the discrepancies in the predictions of the respective studies.

In addition to the theoretical studies, Handwerger et al. (1999) measured *in situ* $^{14}$C concentrations in carbonate deposits (limestone bedrock and tufa) from well-preserved Provo-level shoreline features associated with Pleistocene Lake Bonneville, Utah, to calibrate *in situ* $^{14}$C spallogenic production rates in calcite. The late Pleistocene lake-level history of Lake Bonneville is
well-constrained by traditional radiocarbon dates and has been used for geological calibration of a number of cosmogenic nuclides (Lifton et al., 2015a). *In situ* $^{14}$C measurements in Handwerger et al. (1999) were reduced according to standard methods for radiocarbon in organic materials, but Hippe and Lifton (2014) subsequently developed comprehensive data reduction procedures specifically for *in situ* $^{14}$C. Unfortunately, Handwerger et al. (1999) do not present full details of their analytical results and calculations – we thus cannot correct their data to current standards using the Hippe and Lifton (2014)





protocols. If we assume such corrections would be small relative to the resulting *in situ* $^{14}$C concentrations in their calibration samples, neglecting three anomalous results, and using the age of initial Provo shoreline formation from Lifton et al. (2015a) of 18.3 ± 0.3 cal ka BP, their mean *in situ* $^{14}$C concentration is (3.75 ± 0.26) x $10^5$ atoms g$^{-1}$ CaCO$_3$. This corresponds to a local production rate of ca. 51 atoms g$^{-1}$ yr$^{-1}$. In contrast, the theoretical local production rate calculated with our software framework is ca. 43.9 atoms g$^{-1}$ yr$^{-1}$, ~15% lower than the derived local production rate. In addition, the predicted value normalized to $P_{Qcal}$

yields 37.5 atoms g$^{-1}$ yr$^{-1}$, 27% lower than Handwerger et al. (1999). Given the uncertainties in the uncorrected Handwerger et al. (1999) dataset, and the suggested uncertainties in our method, we find reasonable agreement between our production rate estimates and that of Handwerger et al. (1999).

## 5 Conclusions

As a first step in exploring potential applications of *in situ* $^{14}$C to quartz-poor or fine-grained rock types, we have extended

the functionality of the MATLAB®-based LSDn nuclide-specific scaling framework (Lifton et al., 2014; Lifton, 2016) to estimate spallogenic production of *in situ* $^{14}$C in rock and mineral compositions other than pure quartz at sites of interest. We account for compositionally dependent production by using measured and modelled nucleon excitation functions for target elements in major element oxide analyses (e.g., XRF), in concert with secondary cosmic-ray differential fluxes per Lifton et al. (2014). The ratio of resulting theoretical compositionally dependent *in situ* $^{14}$C production rates to the corresponding theoretical

quartz production rate are then multiplied by the geologically calibrated production rate in quartz, placing the theoretical production rates in a calibrated context. Exploring a broad range of mineral and rock compositions indicates production is dominated by oxygen spallation as expected (>90% at SLHL), but with a general decrease in total production rate with more mafic (particularly Fe-rich) compositions. Although this study confirms previous work identifying Si, Mg, and Al as important targets, we also find for the first time that Na appears to contribute significantly. Future nucleon excitation function

measurements, particularly for Na reactions, should improve the robustness of this software tool further. This framework is thus an important initial step forward in applying *in situ* $^{14}$C to a broader array of landscapes.

## Code availability

The MATLAB® scripts referenced in this manuscript are available at https://github.com/nlifton/CD14C. A permanent DOI will be provided upon acceptance of this manuscript.

## Author contributions


The study was conceived by NL and AK. AK and NL developed the MATLAB® scripts. Manuscript was written by AK and NL.

## Competing interests

The authors declare no competing interests.



**Acknowledgements**

NL received support from the U.S. National Science Foundation (NSF) award EAR-1560658. AK acknowledges support from a

Purdue Research Foundation Ross Fellowship/Assistantship.

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





**Figures**

**Figure 1:** Measured (Reedy, 2013) (top panels) and modelled (bottom panels) neutron and proton reaction excitation functions for *in situ* [14]C production from various targets. Note that modelled predictions for [23]Na (JENDL/HE-2007; Fukahorit et al., 2002; Watanabe et al., 2011)
suggest the highest production of all nuclides considered.

**Figure 2**: Modelled neutron (top) and proton (bottom) cross-sections for [23]Na from JENDL/HE-2007 (Na$_J$, solid line) compared to the spliced TENDL-2019 at energies ≤ 0.2 GeV and JENDL/HE-2007 > 0.2 GeV ([23]Na$_{TJ}$, dashed line). Differential neutron and proton fluxes at SLHL (Sato et al., 2008) are plotted in their respective panes to illustrate the combined effect of excitation function and flux on *in situ* [14]C production.

**Tables**

**Table 1:** Oxide compositions of selected silicate minerals (Barthelmy, 2014) and rock types (Parker, 1967) used to calculate number densities.

**Table 2**: Predicted modern *in situ* [14]C spallogenic production rates (atoms g$^{-1}$ y$^{-1}$) at SLHL from neutrons and protons in minerals and rock types considered, both theoretical (P$_{CDpred}$) and normalized to calibrated production in quartz (P$_{CD}$) using the gridded $R_C$ record of Lifton (2016).

**Table 3:** Percentage of total modern *in situ* [14]C production at SLHL by element for each mineral and rock type considered


**Table 4**: Predicted modern production rates at SLHL for neutron spallation from [16]O derived from secular equilibrium concentrations ($N_{SE}$) at ca. 20-cm depth for different rock types (Fabryka-Martin, 1988) compared to our software framework. Note that these estimates are not normalized relative to $P_{Qcal}$, for straightforward comparison to Fabryka-Martin's (1988) predictions.


**Table 5:** Neutron-only SLHL production based on Masarik (2002; $P_{M02}$) theoretical predictions for compositions considered in this work, compared to modern SLHL neutron-only production predicted here (also see Table 2). Note that these estimates are not normalized relative to $P_{Qcal}$, to enable direct comparison to Masarik's (2002) predictions.






## Tables

**Table 1:**

| Mineral | Composition | SiO$_2$ | TiO$_2$ | Al$_2$O$_3$ | FeO | Fe$_2$O$_3$ | MnO | MgO | CaO | Na$_2$O | K$_2$O | P$_2$O$_5$ | LOI[2] |
|---|---|---|---|---|---|---|---|---|---|---|---|---|---|
| *Quartz* | SiO$_2$ | 100 | - | - | - | - | - | - | - | - | - | - | - |
| *Albite* | NaAlSi$_3$O$_8$ | 52.37 | - | 29.62 | - | - | - | - | - | 18.01 | - | - | - |
| *Anorthite* | CaAl$_2$Si$_2$O$_8$ | 43.19 | - | 36.64 | - | - | - | - | 20.16 | - | - | - | - |
| *Orthoclase* | KAlSi$_3$O$_8$ | 64.76 | - | 18.32 | - | - | - | - | - | - | 16.92 | - | - |
| *Forsterite* | Mg$_2$SiO$_4$ | 42.71 | - | - | - | - | - | 57.30 | - | - | - | - | - |
| *Fayalite* | Fe$_2$SiO$_4$ | 29.49 | - | - | 70.51 | - | - | - | - | - | - | - | - |
| *Wollastonite* | Ca$_2$Si$_2$O$_6$ | 51.72 | - | - | - | - | - | - | 48.28 | - | - | - | - |
| *Augite*[1] | (Ca,Na)(Mg,Fe,Al,Ti)(Si,Al)$_2$O$_6$ | 48.30 | 3.38 | 8.63 | 6.08 | - | - | 15.35 | 21.35 | 1.31 | - | - | - |
| *Ferrosilite* | Fe$_2$Si$_2$O$_6$ | 45.54 | - | - | 54.46 | - | - | - | - | - | - | - | - |
| *Enstatite* | Mg$_2$Si$_2$O$_6$ | 59.85 | - | - | - | - | - | 40.15 | - | - | - | - | - |
| *Calcite*[2] | CaCO$_3$ | - | - | - | - | - | - | - | 56.03 | - | - | - | 43.97 |
| *Dolomite*[2] | CaMg(CO$_3$)$_2$ | - | - | - | - | - | - | 21.86 | 30.41 | - | - | - | 47.73 |
| **Rock type**[3] | | | | | | | | | | | | | |
| *Ultramafic* | – | 40.64 | 0.05 | 0.66 | - | 14.09 | 0.19 | 42.94 | 0.98 | 0.77 | 0.04 | 0.04 | - |
| *Basalt* | – | 51.34 | 1.50 | 16.55 | - | 12.24 | 0.26 | 7.46 | 9.40 | 2.62 | 1.00 | 0.32 | - |
| *Hi-Ca Granite* | – | 67.16 | 0.57 | 15.49 | - | 4.23 | 0.07 | 1.56 | 3.54 | 3.83 | 3.04 | 0.21 | - |
| *Low-Ca Granite* | – | 74.22 | 0.20 | 13.60 | - | 2.03 | 0.05 | 0.27 | 0.71 | 3.48 | 5.06 | 0.14 | - |
| *Granodiorite* | – | 69.09 | 0.57 | 14.55 | - | 3.86 | 0.08 | 0.93 | 2.21 | 3.73 | 4.02 | 0.16 | - |

[1] Assumed empirical composition of augite (Barthelmy, 2014): (Ca$_{0.9}$Na$_{0.1}$)(Mg$_{0.9}$Fe$^{2+}_{0.2}$Al$_{0.4}$Ti$_{0.1}$)Si$_{1.9}$O$_6$

[2] LOI = Loss on ignition. Assumed to be entirely CO$_2$ for carbonates, used in oxygen number density calculation

[3] Compositions from Parker (1967)





**Table 2**

| Mineral | Neutron $P_{CDpred}$ at g$^{-1}$ y$^{-1}$ | Proton $P_{CDpred}$ at g$^{-1}$ y$^{-1}$ | Total $P_{CDpred}$ at g$^{-1}$ y$^{-1}$ | $P_{CD}$ at g$^{-1}$ y$^{-1}$ | % Diff $P_{CD}$ vs. $P_{Qcal}$ |
|---|---|---|---|---|---|
| Quartz | 15.37 | 0.47 | 15.84 | 13.53 | 0.0 |
| Albite | 15.55 | 0.48 | 16.04 | 13.70 | 1.2 |
| Albite[1] | 14.74 | 0.48 | 15.22 | 13.00 | -4.0 |
| Anorthite | 13.43 | 0.42 | 13.85 | 11.80 | -12.6 |
| Orthoclase | 13.35 | 0.42 | 13.77 | 11.73 | -13.1 |
| Forsterite | 13.66 | 0.46 | 14.12 | 12.03 | -10.9 |
| Fayalite | 9.07 | 0.28 | 9.35 | 7.97 | -41.0 |
| Wollastonite | 11.85 | 0.36 | 12.21 | 10.41 | -22.9 |
| Augite | 13.28 | 0.42 | 13.70 | 11.67 | -13.6 |
| Ferrosilite | 10.46 | 0.32 | 10.78 | 9.18 | -32.0 |
| Enstatite | 14.17 | 0.46 | 14.64 | 12.50 | -7.6 |
| Calcite | 13.55 | 0.38 | 13.93 | 11.87 | -12.1 |
| Dolomite | 14.96 | 0.44 | 15.41 | 13.13 | -2.8 |
| **Rock** | | | | | |
| Ultramafic | 13.11 | 0.43 | 13.54 | 11.56 | -14.5 |
| Basalt | 13.72 | 0.43 | 14.15 | 12.08 | -10.7 |
| Hi-Ca Granite | 14.30 | 0.44 | 14.75 | 12.59 | -6.9 |
| Low-Ca Granite | 14.52 | 0.45 | 14.97 | 12.79 | -5.5 |
| Granodiorite | 14.27 | 0.44 | 14.71 | 12.57 | -7.1 |


[1]Production is calculated using the spliced TENDL-2019 and JENDL/HE-2007 proton and neutron excitation functions (Na$_{TJ}$ in text)





**Table 3**

|  | O | Si | Ti | Al | Fe$^{2+}$ | Fe$^{3+}$ | Mn | Mg | Ca | Na | K | P |
|---|---|---|---|---|---|---|---|---|---|---|---|---|
| **Minerals** | | | | | | | | | | | | |
| *Quartz* | 97.5 | 2.5 | - | - | - | - | - | - | - | - | - | - |
| *Albite* | 88.09 | 1.29 | - | 1.63 | - | - | - | - | - | 12.99 | - | - |
| *Albite[1]* | 88.62 | 1.36 | - | 1.72 | - | - | - | - | - | 8.30 | - | - |
| *Anorthite* | 96.37 | 1.23 | - | 2.33 | - | - | - | - | 0.07 | - | <0.01 | - |
| *Orthoclase* | 96.89 | 1.85 | - | 1.17 | - | - | - | - | - | - | 0.08 | - |
| *Forsterite* | 93.44 | 1.19 | - | - | - | - | - | 5.37 | - | - | - | - |
| *Fayalite* | 98.14 | 1.24 | - | - | - | 0.61 | - | - | - | - | - | - |
| *Wollastonite* | 98.16 | 1.67 | - | - | - | - | - | - | 0.18 | - | - | - |
| *Augite* | 95.35 | 1.39 | <0.01 | 0.56 | - | <0.01 | - | 1.48 | 0.07 | 1.11 | - | - |
| *Ferrosilite* | 97.93 | 1.66 | - | - | - | 0.41 | - | - | - | - | - | - |
| *Enstatite* | 94.76 | 1.61 | - | - | - | - | - | 3.63 | - | - | - | - |
| *Calcite* | 99.82 | - | - | - | - | - | - | - | 0.18 | - | - | - |
| *Dolomite* | 98.19 | - | - | - | - | - | - | 1.74 | 0.07 | - | - | - |
| **Rock type** | | | | | | | | | | | | |
| *Ultramafic* | 93.84 | 1.18 | <0.01 | 0.04 | 0.08 | <0.01 | <0.01 | 4.20 | <0.01 | 0.66 | <0.01 | <0.01 |
| *Basalt* | 94.60 | 1.43 | <0.01 | 1.08 | 0.07 | <0.01 | <0.01 | 0.70 | 0.03 | 2.14 | <0.01 | <0.01 |
| *Hi-Ca Granite* | 94.09 | 1.79 | <0.01 | 1.01 | 0.02 | <0.01 | <0.01 | 0.14 | 0.01 | 3.00 | 0.01 | <0.01 |
| *Low-Ca Granite* | 94.50 | 1.95 | <0.01 | 0.89 | 0.01 | <0.01 | <0.01 | 0.02 | <0.01 | 2.69 | 0.02 | <0.01 |
| *Granodiorite* | 94.22 | 1.85 | <0.01 | 0.95 | 0.02 | <0.01 | <0.01 | 0.08 | 0.01 | 2.93 | 0.02 | <0.01 |


[1]Production is calculated using the spliced TENDL-2019 and JENDL/HE-2007 proton and neutron excitation functions (Na$_{TJ}$ in text)

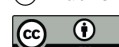



**Table 4**


| Rock Type | Depth $(m)$[1] | density | $N_{SE}$ $(at\ g^{-1})$[1] | $P_{16O\text{-}FM}$[1] $(at\ g^{-1}y^{-1})$ | $P_{16O}$[2] $(at\ g^{-1}y^{-1})$ |
|---|---|---|---|---|---|
| *Ultramafic* | 0.18 | | 135706 | 16.4 | 9.0 |
| *Basalt* | 0.18 | | 132621 | 16.0 | 9.3 |
| *Hi-Ca Granite* | 0.19 | | 148043 | 17.9 | 9.7 |
| *Low-Ca Granite* | 0.19 | | 151127 | 18.3 | 9.9 |
| *Limestone* | 0.19 | | 151127 | 18.3 | 10.1 |

[1]Data from Fabryka-Martin (1988), assumes SLHL production rate from oxygen in Yokoyama et al. (1977)
[2]Data from this study assuming only production from neutron spallation of O and an attenuation length of 160 g cm$^{-2}$

**Table 5**


| Mineral | $P_{M02}$ $(at\ g^{-1}\ y^{-1})$ | $P_{CDn}$ $(at\ g^{-1}\ y^{-1})$ |
|---|---|---|
| *Quartz* | 18.72 | 15.37 |
| *Albite* | 19.99 | 15.56 |
| *Anorthite* | 16.25 | 13.43 |
| *Orthoclase* | 16.20 | 13.35 |
| *Forsterite* | 16.43 | 13.66 |
| *Fayalite* | 11.06 | 9.07 |
| *Wollastonite* | 14.42 | 11.85 |
| *Augite* | 15.91 | 13.28 |
| *Ferrosilite* | 14.85 | 10.46 |
| *Enstatite* | 17.11 | 14.17 |
| *Calcite* | 16.48 | 13.55 |
| *Dolomite* | 18.12 | 14.96 |
| **Rock** | | |
| *Ultramafic* | 15.27 | 13.11 |
| *Basalt* | 15.38 | 13.72 |
| *Hi-Ca Granite* | 17.15 | 14.30 |
| *Low-Ca Granite* | 17.15 | 14.52 |
| *Granodiorite* | 17.14 | 14.27 |





**Figure 1**



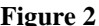

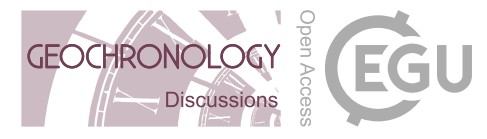

**Figure 2**

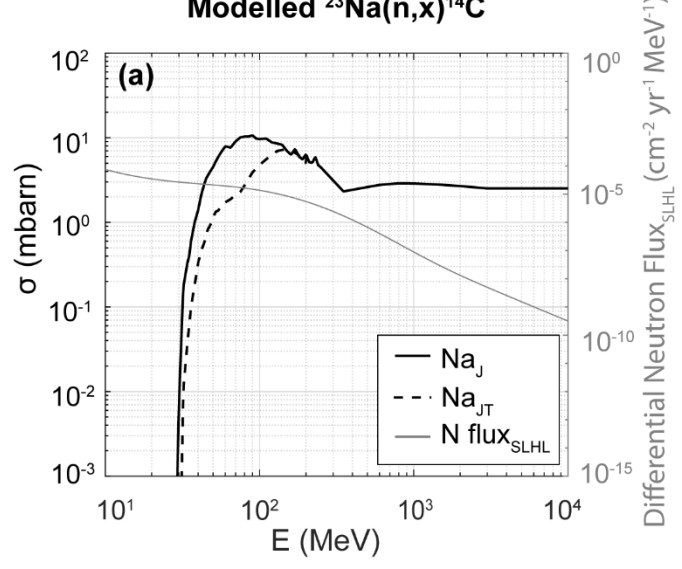
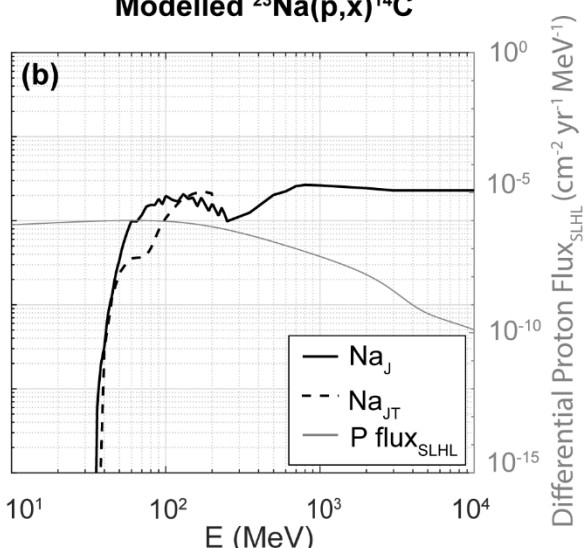