# Peer review of "Technical Note: A software framework for calculating compositionally dependent *in situ* 14C production rates"

_Geochronology, 2022_

## Author Comment (AC2)

[Figure]

**Figure 3:** SLHL production of minerals (left) and rocks (right) compared to quartz (dashed grey line). The minerals and rocks are colored for what element contributes the highest proportion of production after oxygen and silica.

---

## Author Response (AR1)

**Reto Trappitsch**

In the manuscript: "A software framework for calculating compositionally dependent in situ 14C production rates", Koester and Lifton present new results for calculating cosmogenic 14C production rates for various minerals other than quartz. These calculations are based on previous models and represent an extension that allows experimentalists to calculate exposure histories for quartz-poor samples.

These new production rates significantly broaden the applicability of exposure age dating via the cosmogenic 14C to a variety of minerals other than quartz. The results present a major contribution to the field of geochronology and are therefore ideally suited for publication in this journal. I highly recommend publication of this manuscript, however, would like to propose several clarifications / edits.

Major Comments:

================

You present production rates based on various minerals and give the composition of these minerals. Would it be easier / simpler to present elementary production rates and have the user ultimately calculate the overall production rate in a given mineral based on their specific composition? This could allow a user to easier work with the results from this manuscript. Please feel free to completely ignore this idea, I am not very familiar with the general approach in the field of terrestrial cosmogenic nuclides.

- We appreciate the reviewer's comments, but the calculator already does what is suggested. It takes major elemental oxide data and calculates a theoretical production rate based on the sample composition. The production rate is then normalized to the calibrated production rate in quartz. Future work would include a geologic calibration for each mineral if possible.

Section 3.1, second paragraph: In Reedy (2013), excitation functions for the production of 14C from elementary O and Si are presented and not from 16O and 28Si. Generally, cross-sections are measured using materials with natural isotopic abundances. This makes more sense, since these elements also occur in geological samples of interest in their normal isotopic composition.

- We thank the reviewer for noting the inconsistency in our manuscript. We have changed the wording in our description of Reedy's measurements and updated the figures that show the measured excitation functions, noting that they are on natural isotopic abundances.

For excitation functions from JENDL/HE-2007: Did you take the values for pure isotopic compositions as stated in line 128? I don't expect that your samples contain, e.g.,

isotopically pure 48Ti. Therefore, a Ti(n,X)14C excitation function where all isotopes are included in their terrestrial composition should be used for the calculations.

- The reviewer is correct that we didn't use pure isotopic compositions for modelled excitation functions and we have corrected the statement for clarity. For the modelled excitation functions, we chose to use the most abundant isotope for each element. The three isotopes we choose that aren't 100% are 39K (93% elemental abundance), 40Ca (97% elemental abundance), and 48Ti (73% elemental abundance). Although this excludes other isotopes of K, Ca, and Ti, we note that the proportion of spallation production from each is small for the compositions we chose. For instance, the percentage of production from 39K ranges from 0.1 to <0.01, 40Ca ranges from 0.2 to 0.01, and Ti is <0.001.

Uncertainty determination for JENDL/HE-2007 excitation functions, last paragraph in Section 4.2. You estimate the overall uncertainties for the purely calculated excitation functions at 10-15%. The estimate is based on comparing the JENDL/HE-2007 (calculated, generally GNASH) with TENDL (calculated, TALYS). This comparison is not exactly fair. A better comparison would be to compare calculated excitation functions with measured ones, as, e.g., Broeders et al. (https://doi.org/10.14494/JNRS2000.7.N1), however, you obviously cannot do this for the reactions you are interested in. A better estimate to use for the uncertainty of calculated production rates is given in Reedy (2013), section 3.1, third-last paragraph: "[...] most formulae and codes give cross-sections for an individual nuclide that typically differ from measured ones by factors of ~2 (Ammon et al., 2009)". Your uncertainty estimate seems therefore far too optimistic.

- We agree that our uncertainty estimates are optimistic, and we note in our conclusion that measured excitation functions would greatly improve our code. Apart from 23Na, all the modelled reaction cross sections have little impact on the overall production rate. The percentage of production of 55Mn, 40Ca, 39K and 31P range from <0.001 to 0.2 for our range of compositions. Even if reaction cross sections are off by a factor of 2, the impact to overall production is small. For instance, if we doubled the percentage of Ca production for Wollastonite, it would only increase to 0.4 %.

Minor Comments:

===============

Section 3.1, first paragraph: This paragraph contains quite a lot of information that is not understandable without reading Lifton et al. (2014) first. You are already describing what goes into the model in a very good way in Section 2.1. For this paragraph, it would be good if you could explain all abbreviations (LSDn, PARMA, SHA.DIF.14k - if this is an abbreviation). Furthermore, could you provide some detail on what the gridded R_C and dipolar R_CD models of Lifton et al. (2016) are?

- We acknowledge that we skimmed over the gritty details of Lifton et al. (2014) and believe that readers can go to the original publication for full details of the LSDn methodology. We have added a sentence in section 3.1 to refer readers to the original publication for details. We expect that future readers will be familiar with the terms "LSDn" because it is a typical scaling method (along with St and Lm, as seen in the online Cronus Cal v3 we reference in text).
- We have included some minors edits in line 115 for clarification.

Figure 1: For the measured curves, it might be good to present the measurements as symbols, in order to easier distinguish between interpolated and measured values.

- We appreciate the comment to clarify the figure and have added symbols to make it easier to distinguish measured values from interpolated values.

**Irene Schimmelpfennig:**

This manuscript reports theoretic production rates of in situ cosmogenic 14C in mineral and rock phases of various compositions, estimated from a specifically developed software framework.

Given the potential need for knowledge of 14C production rates in minerals and rocks other than quartz in future surface exposure dating studies, this manuscript is well suited for publication in Geochronology.

It is very well and clearly written. I suggest a few minor clarifications that should be addressed.

Lines 55-56: It could be good to clarify that the extraction procedures from mineral/rock phases other than quartz also still needs to be developed before these materials can be envisioned for geologic applications.

- We thank the review for their comment. We have clarified this point in the manuscript at the end of our introduction (line 63-64)

Lines 80-81: This sentence is unclear: does "well-constrained" refer to the exposure history? Natural variability of what? I don't understand the point of the sentence.

- Yes, "well-constrained" does refer to the exposure history. We have edited the sentence for clarity.

Lines 81-82: The focus on quartz is also due to the fact that extraction procedures for other minerals or lithologies have not yet been developed or validated.

- We note that the first extraction techniques for in situ 14C were from whole rock samples (e.g., Jull et al., 1992; 1994) in the introduction and were abandoned in favor of the simpler system of quartz. (lines 55-56).

Line 149: Were elevation differences between individual samples at each site insignificant? Or were the concentrations corrected for them?

- In this 14C Cronus global calibration, there are saturated samples along elevation transects that span a broad altitudinal range at a particular latitude. Those samples are incorporated into the estimation of the SLHL production rate. In addition, there are calibration sites with multiple samples that span limited altitudinal ranges.

Lines 153-154: What is the calibrated value generated by the UWv3 calculator?

- The output value generated by the UWv3 calculator is 0.868 and used a dipolar RC. This value is the fitting parameter that is multiplied by the reference production rate for 14C, 15.8 atoms/g. This yields a value of 13.7, which is the same value we get for dipolar RC in the manuscript (see line 154) and already mentioned in text (line 156).

Lines 159-165: This should be simplified by saying that you calculate a correction factor P_Qcal/P_Qref , which gives 0.854 and which you multiply by the P_CDpred of all other tested mineral and rock phases. However, how reliable is this correction for other compositions, which are associated with other excitation functions than quartz?

- We prefer to keep the equation as presented because the P_CDpred and the P_Qref are both theoretical and we multiply that by the geologic calibration of quartz. Therefore, we keep theoretical components together and separate from the geologic component.
- The correction is as reliable as possible given our current knowledge of the scaling of fluxes, reaction cross sections, and geologic calibration. As better reaction cross sections are measured and mineral specific geologic production rate calibrations are conducted, these corrections will likely improve.

Lines 237-238: Would it be possible to list the elemental 14C production rates, for direct comparison with those given in the Masarik (2002) abstract? This is also what is commonly done for the highly composition dependent 36Cl production.

- We thank the reviewer for the suggestion and have included a list of production from each element within the text (shown as atoms g-element$^{-1}$ yr$^{-1}$) as equation 4.

Related to this, I suggest you should clarify whether or not the software also calculated production rates for compositions that differ from those considered here theoretically.

- The software framework can take in any XRF elemental analysis for any rock or mineral type. The user can input any elemental oxide percent from any location and calculate a theoretical production rate. We have added a sentence to line 143 for clarity.

Caption of Table 1: Shouldn't this be "Oxide compositions… and accordingly calculated number densities"? (It should be clarified what the numbers are.)

- The table lists the percentage of oxide compositions for each mineral and rock. These percentages are used to calculate the number densities as in equation 2 (line 139). We have edited the caption for clarity.

- While we were at a conference, the reviewer (Dr. Schimmelpfennig) suggested in person to include a figure of the production from minerals and rocks from a presentation about this work. It is now Figure 3.